# Neurocognitive Profile of the Post-COVID Condition in Adults in Catalonia—A Mixed Method Prospective Cohort and Nested Case–Control Study: Study Protocol

**DOI:** 10.3390/vaccines10060849

**Published:** 2022-05-26

**Authors:** Rosalia Dacosta-Aguayo, Noemí Lamonja-Vicente, Carla Chacón, Lucia Amalía Carrasco-Ribelles, Pilar Montero-Alia, Anna Costa-Garrido, Rosa García-Sierra, Victor M. López-Lifante, Eduard Moreno-Gabriel, Marta Massanella, Josep Puig, Jose A. Muñoz-Moreno, Lourdes Mateu, Anna Prats, Carmina Rodríguez, Maria Mataró, Julia G. Prado, Eva Martínez-Cáceres, Concepción Violán, Pere Torán-Monserrat

**Affiliations:** 1Unitat de Suport a la Recerca Metropolitana Nord, Institut Universitari d’Investigació en Atenció Primària Jordi Gol (IDIAP Jordi Gol), 08303 Mataró, Spain; rdacosta@ub.edu (R.D.-A.); nlamonja@idiapjgol.info (N.L.-V.); cchaconv.mn.ics@gencat.cat (C.C.); lcarrasco@idiapjgol.info (L.A.C.-R.); pmontero.bnm.ics@gencat.cat (P.M.-A.); acosta@igtp.cat (A.C.-G.); rgarciasi.mn.ics@gencat.cat (R.G.-S.); vmlopezli.mn.ics@gencat.cat (V.M.L.-L.); emorenoga.mn.ics@gencat.cat (E.M.-G.); crodriguezp.bnm.ics@gencat.cat (C.R.); ptoran.bnm.ics@gencat.cat (P.T.-M.); 2Comparative Medicine and Bioimaging Center (CMCiB), Germans Trias i Pujol Research Institute, 08916 Badalona, Spain; jpuigmd@gmail.com; 3Department of Clinical Psychology and Psychobiology, University of Barcelona, 08035 Barcelona, Spain; mmataro@ub.edu; 4Direcció d’Atenció Primària Metropolitana Nord Institut Català de Salut, 08916 Mataró, Spain; 5Multidisciplinary Research Group in Health and Society GREMSAS (2017 SGR 917), 08007 Barcelona, Spain; 6Centre d’Atenció Primària La Riera (Mataró 1), Institut Català de la Salut, 08302 Barcelona, Spain; 7Nursing Department, Faculty of Medicine, Universitat Autònoma de Barcelona, 08193 Barcelona, Spain; 8Palau-Solità Healthcare Centre, Palau-Solità Plegamans Institut Català de la Salut, 08124 Barcelona, Spain; 9Department of Social Psychology, Universitat Autònoma de Barcelona, Cerdanyola de Vallès, 08193 Bellaterra, Spain; 10IrsiCaixa-AIDS Research Institute and Germans Trias i Pujol Health Research Institute (IGTP), Can Ruti Campus, 08916 Badalona, Spain; mmassanella@irsicaixa.es (M.M.); lmateu.germanstrias@gencat.cat (L.M.); jgarciaprado@irsicaixa.es (J.G.P.); 11Centro de Investigación Biomédica en Red de Enfermedades Infecciosas (CIBERINFEC), Instituto de Salud Carlos III (ISCIII), 28029 Madrid, Spain; 12University of Vic-Central University of Catalonia (UVic-UCC), 08500 Vic, Spain; 13Infectious Diseases Department, Fight against AIDS Foundation (FLS), Germans Trias i Pujol Hospital, Can Ruti Campus, 08916 Badalona, Spain; jmunoz@flsida.org (J.A.M.-M.); aprats@flsida.org (A.P.); 14Facultat de Psicologia i Ciències de l’Educació, Universitat Oberta de Catalunya (UOC), 08035 Barcelona, Spain; 15Germans Trias i Pujol Hospital, Universitat Autònoma de Barcelona, 08916 Badalona, Spain; 16Sant Fost de Campcentelles Healthcare Centre, Sant Fost de Campcentelles, Institut Català de la Salut, 08105 Barcelona, Spain; 17Germans Trias i Pujol Research Institute, Universitat Autònoma de Barcelona, 08916 Badalona, Spain; 18Immunology Department, FOCIS Center of Excellence—Universitat Autònoma de Barcelona, 08193 Cerdanyola del Vallès, Spain; emmartinez.germanstrias@gencat.cat; 19Immunology Division, Laboratori Clinic Metropolitana Nord (LCMN), Hospital Universitari Germans Trias i Pujol, 08916 Badalona, Spain; 20Department of Medicine, Faculty of Medicine, Universitat Autònoma de Barcelona, 08193 Bellaterra, Spain; 21Universitat Autònoma de Barcelona, 08193 Bellaterra, Spain; 22Department of Medicine, Faculty of Medicine, Universitat de Girona, 17003 Girona, Spain

**Keywords:** post-COVID condition, cognitive symptoms, neuropsychologic symptoms, quality of life, retinography, posturography, magnetic resonance imaging, immune response, inflammation

## Abstract

The diagnosis of the post-COVID condition is usually achieved by excluding other diseases; however, cognitive changes are often found in the post-COVID disorder. Therefore, monitoring and treating the recovery from the post-COVID condition is necessary to establish biomarkers to guide the diagnosis of symptoms, including cognitive impairment. Our study employs a prospected cohort and nested case–control design with mixed methods, including statistical analyses, interviews, and focus groups. Our main aim is to identify biomarkers (functional and structural neural changes, inflammatory and immune status, vascular and vestibular signs and symptoms) easily applied in primary care to detect cognitive changes in post-COVID cases. The results will open up a new line of research to inform diagnostic and therapeutic decisions with special considerations for cognitive impairment in the post-COVID condition.

## 1. Introduction

As of 5 February 2022, COVID-19 has affected more than 489 million people globally and caused more than 6 million deaths (https://coronavirus.jhu.edu/map.html accessed on 2 April 2022). The number of severe cases, including those with hospitalization and death, has been drastically reduced by administering more than 9.8 billion SARS-CoV-2 vaccines worldwide [1]. Although vaccines limit severity, it has been shown that they do not prevent the transmission of the virus [2]. It is especially the case for the Omicron variant, which has already led to an increase in post-vaccine infections [2]. A recent systematic review [3] indicates that infection (or reinfection) risks vary between 2 and 16%.

The large volume of new patients and reinfections that have appeared in this latest wave also increases the concern about the foreseeable growth of cases with prolonged symptoms of what has been called the post-COVID condition (PCC) [4]. The PCC does not seem to be related to the severity of the initial infection because it can affect both patients who have had mild, even asymptomatic COVID-19, patients with more severe cases, and patients who have been hospitalized [5]. It occurs in people of all ages, being more common in middle-aged women [6], with a severe impact on the quality of life in the family, social, and work environment.

In late 2021, the WHO proposed PCC as the condition that appears in people with a history of probable or confirmed SARS-CoV-2 infection, usually three months after onset, with symptoms lasting at least two months and which cannot be explained by an alternative diagnosis (WHO, 2021). The most common symptoms are fatigue, shortness of breath, and cognitive dysfunction, but other symptoms may also appear that affect the daily lives of those suffering from the disease. (WHO, 2021). The prevalence of persistent symptoms is between 10% and 15% [7]. 

Persistent neurological, neuropsychological, and neuropsychiatric symptoms, which affect people’s cognition, have also been described following SARS-CoV-2 infection, and a new corresponding term “NeuroCOVID” has also been introduced [8]. 

Subjective cognitive complaints appear to be the most common symptoms reported by patients after SARS-CoV-2 recovery [9,10]. Some authors have found that those subjective cognitive complaints are associated with objective cognitive deficits, such as attention (brain fog), coding of verbal memory, long-term verbal memory, working memory, executive function, praxis, and verbal fluency [11,12,13,14,15,16,17,18,19]. Others, however, have not found this association [20,21,22]. A significant limitation of these studies is that they focus on the cognitive impact of SARS-CoV-2 in patients who have been hospitalized, and there is a lack of data on patients with mild forms of infection. Understanding the effects of SARS-CoV-2 on cognition is paramount in implementing appropriate cognitive prevention and rehabilitation in these patients.

SARS-CoV-2-induced neurological complications include headache and neuralgia, myositis, Guillain Barré syndrome, and movement difficulties due to basal ganglia dysfunctions (flutter, tremors, myoclonus, and ataxia) [23,24,25,26,27,28], and a case of recurrent SARS-CoV-2 with neuro vestibular symptoms has also been reported [29,30]. Cerebrovascular disorders, such as optic neuritis, retinal hemorrhage, and cotton blemishes, have also been registered, with the most common manifestation being retinal vein occlusion with associated macular edema [31,32]. It is known that changes in the retinal vessels can serve as a biomarker of vascular changes in the whole body, especially in the cerebral vessels, because retinal vessels share the same exposure to thromboembolic complications [33]. Several reports have shown signs of vascular disorders in the retina of patients with SARS-CoV-2 [31,32,34]. Retinography is a technique for analyzing the composition and morphology of the retinal vascular tree. Retinography is available in primary care settings and can be very useful in diagnosing PCC-associated cerebrovascular injury in combination with magnetic resonance imaging (MRI). Our group has found associations between the markers of retinal microvasculature observed in mydriatic camera retinography and the presence of silent cerebrovascular disease [35].

Fatigue manifested as exhaustion is another complaint, found mainly in women [36] and has become increasingly relevant in patients with SARS-CoV-2 [37,38,39]. This symptom, along with cognitive impairment, had compromised the quality of life of these patients, as reported in a study in which 45.2% of patients required a reduced workload compared to their condition before PCC, and in which 22.3% of patients had not yet returned to work [40]. The underlying neuronal substrate of fatigue is mainly unknown, although hypothalamic changes, also observed in chronic fatigue syndrome and myalgic encephalitis [41], may be responsible [42].

SARS-CoV-2 is also known to cause neuropsychiatric complications [43] and exacerbation of previous psychological disorders, such as anxiety and depression [44,45,46,47]. The causes of these manifestations are multifactorial and different mechanisms have been described, such as direct invasion of the virus into the brain (encephalitis/cerebral hypoxia) [48], the consequence of a deregulated immune response and a virus-induced cytokine storm [49,50], as well as physical isolation, psychosocial impact, and social stigma [51]. A systematic review reported post-traumatic stress disorder (PTSD), irritability, anxiety, insomnia, depression, obsessive-compulsive disorder (OCD), and paranoia as the most prevalent symptoms [51].

Neuropathological studies in dead patients have revealed edema, partial neuronal disease and the involvement of the white matter [52]. Several pathophysiological and metabolic mechanisms have been proposed to explain these changes. First, the virus could infect the brain through the angiotensin 2-converting enzyme receptor (ACE2) [53] because microglia and neurons have this receptor [54]. Second, hypoxia and coagulation dysfunction caused by SARS-CoV-2 can also affect the brain [53]. Third, SARS-CoV-2 can also cause direct neuronal dysfunctions if it can reach the brain [55]. Fourth, the hyperinflammatory state associated with SARS-CoV-2 infection may lead to glial activation and demyelination of the CNS [56] because demyelination has been detected along with virus particles in the brain tissue of people infected with SARS-CoV-2 [56]. 

A fluorodeoxyglucose positron emission tomography (FDG-PET) study with COVID patients [57] described dysfunction in the cingulate cortex, an anatomical region involved in emotions, memory, depression, and decision making [58]. Several studies have observed high levels of various cytokines, especially IL-6, TNF-α, IL-1b, and C-reactive protein. IL-6 and ferritin are involved in many psychiatric diseases, such as depression, PTSD, and OCD [59,60,61,62], consistent with the data on patients with SARS-CoV-2 [63]. These inflammatory disorders are also evidenced by structural and functional magnetic resonance imaging studies that show the disruption of fractional anisotropy and axial diffusivity and functional alterations of the hippocampus, part of a subcortical network involved in the responses to anxiety and stress [15]. The use of psychological tests to measure depression, anxiety, personality disorders, and insomnia, along with the use of neuroimaging techniques, allow for a better understanding of the cognitive well-being of patients with PCC and facilitate the development of psychological therapies and neuropsychological tools to improve the quality of life of patients with PCC.

The broad spectrum of PCC’s psychological, cognitive, and neurological symptoms results in an extensive study, such as the one proposed, which allows for the analyses of the causes, signs, and symptoms to achieve a mechanistic understanding of the PCC and their subsequent treatment. Such studies can identify biomarkers, including functional and structural connectivity, inflammatory, immunological, retinal vessel involvement, and vestibular impairment that can inform health services, especially the primary care, guide the diagnosis and monitor the treatment. 


**Hypothesis**


When comparing the COVID-19PNC group with patients with PCC and with no neurocognitive deficits at all and during the study period (COVID-19PnoNC), with patients who recovered from COVID-19 (COVID-19R) and with a group of persons who never had COVID-19 and without SARS-CoV-2 infection during the study period considered as healthy controls (NoCOVID-19), we expect to find alterations in the different biomarkers’ functional and structural connectivity, inflammatory, immunological, retinal vessel involvement, and vestibular impairment in the COVID-19PNC group concerning the control groups. 


**Objectives**



**Main Aim**


The main aim is to characterize the neurocognitive PCC by analyzing the relationships between the accompanying functional and structural neural changes, inflammatory and immune status, vascular and vestibular involvement, and their impact on the activities of daily living. 


**Specific Aims**


The specific aims of this study include the investigation of the changes presented within the persistent neurocognitive deficits in PCC (COVID-19PNC) concerning the other three control groups, which are as follows: PCC patients with no neurocognitive deficits (COVID-19PnoN), patients recovered from COVID-19 (COVID-19R), and healthy controls that had no COVID-19 (NoCOVID-19). Specifically, we will carry out the following steps: Analyze the structural and functional changes in the brain, the eye, and the vestibular system in COVID-19 patients, employing functional and structural magnetic resonance imaging (MRI) studies, retinography, and posturography;Analyze the neuropsychological status, emotional state, and quality of life using neuropsychological questionnaires;Analyze different inflammatory and immune biomarkers produced as a response to SARS-CoV-2;Investigate the relationship between factors and biomarkers acquired in 1–3.Describe the impact of persistent symptoms in the context of daily life through individualized interviews and four focus groups (the same ones as for the quantitative part).

## 2. Materials and Methods

### 2.1. Design

The study uses concurrent mixed quantitative and qualitative methodology.

The quantitative methodology is applied to a prospective cohort study among patients following the start of the SARS-CoV-2 pandemic in Spain (March 2020) until the earliest date of symptoms of PCC appeared on 31 December 2024. Qualitative methods complement the quantitative data, providing a broader understanding of the phenomenon. All projects have been approved by the research ethics committee of the University Institute for Research in Primary Care (IDIAPJGol, 21/220-P).

A nested case–control study was also conducted to estimate the association between the alteration in the biomarkers (functional and/or structural neural changes, inflammatory and immune status, vascular and vestibular) and risk of COVID-19PNC.

Patients will be matched at baseline: 120 cases and 40 control cases for each control group. The basis group for matching is the case group. Four independent groups are all matched by age and sex (3:1:1:1). In the case and control group, we considered people from 25 to 65 years old.

The case group (COVID-19PNC) includes 120 people suffering from PCC and with persistent neurocognitive symptoms, as defined by WHO in 2021.
-Control group 0 (NoCOVID-19) includes 40 people who are not infected with SARS-CoV2. -Control group 1 (COVID-19PnoN) includes 40 people suffering from PCC but without any neurocognitive deficit, according to the WHO definition, 2021. -Control group 2 (COVID-19R) includes 40 people who have recovered from SARS-CoV-2 infection, without cognitive impairment or psychiatric symptoms prior to COVID-19 infection (See Figure 1).

The qualitative methodology includes an interpretive phenomenological exploratory study based on semi-structured interviews and four focus groups, with one per group.

### 2.2. Participants, Exposure

The study includes participants ≥ 25 years old, who were admitted to a primary care center or to a hospital care center in the public network of Catalonia from 3 January 2020. This population will be followed-up until 31 December 2024. For this sub-study, the inclusion period is from 1 August 2020 to 31 December 2024. The index data for the COVID-19PNC group (case) is considered the first data, and the following symptoms are defined in the inclusion criteria.

The inclusion criteria are as follows: (a) For the COVID-19PNC group (case), the patients must present at least one of the following symptoms. 1. Present any persistent neurological symptoms of COVID-19 for a period > 12 weeks from the onset of the disease, including recurrent migraines, anosmia, ageusia, dizziness, and vertigo. 2. Present any neuropsychological symptom for a period > 12 weeks from the onset of the disease, including the subjective sensation of impairment of one or more cognitive functions. 3. Present any persistent vestibular or neuropsychiatric symptoms of COVID-19 for a period > 12 weeks from the onset of the disease, including depressive disorders, anxiety, post-traumatic stress disorder. 4. The person must have two or more symptoms from points 1 and 3. Two groups of COVID-19PNC patients will be considered, including persons with 6 to 12 months of symptoms and persons with more than 12 months of symptoms.

(b) For control group 0 (NoCOVID-19), the participants must not be infected with COVID-19, as proved with a negative Rt-PCR and negative nucleocapsid serology; (c) for control group 1 (COVID-19PnoN), the participants must present some persistent symptoms of COVID-19 (i.e., PCC) but no neurocognitive deficit for a period > 12 weeks from the onset of the disease; (d) for control Group 2 (COVID-19R), the participants must have recovered from COVID-19 infection, proved with a positive Rt-PCR and present no persistent symptoms of COVID-19 for a period of > 12 weeks from the onset of the disease.

The exclusion criteria are as follows: 1. cognitive impairment, psychiatric symptoms, such as psychosis, schizophrenia, attention-deficit, history of drug or alcohol abuse, or neurological disease prior to COVID-19 infection; 2. presentation of any condition that prevents or contraindicates MRI; 3.a vital prognosis of < 6 months.

### 2.3. Recruitment

The study will involve the 70 primary care centers in the Northern metropolitan area of Barcelona and the Germans Trias I Pujol University Hospital. The participants will always be evaluated by the same neuropsychologist who is trained and prepared to perform the different diagnostic tests of the study. The test will be standardized according to every participant’s age and years of education. The participants will be all assessed in one center in Mataró. The participants belong to two different cohorts (ProHEpiC_19 and King extension project) to whom we will offer to participate in the new study. They will sign informed consent for the new project.

### 2.4. Sample Size and Sampling Procedure

The quantitative methodology includes accepting an alpha risk of 0.05 and a beta risk of 0.2 in a bilateral contrast; 120 cases with neurocognitive impairment and 40 for each control group are required to detect a difference of 0.26 between the proportions of functional alterations between the groups.

A random sampling of all the participants who meet the inclusion criteria for the case group and the control groups will be performed. CASE participants will be grouped by age and sex. Then, we will randomly select participants from the control groups in a proportion of 3:1:1:1, CASE: CG0:CG1:CG2. 

The qualitative methodology involves the final sample, which will be determined by the saturation of the data, even though the projected sample includes 64 participants between 32 semi-structured interviews and 4 focus groups (from 8 participants each), distributed sequentially in 3 phases (16 interviews exploratory studies, 4 focus groups, and 16 in-depth interviews) to ensure the principles of qualitative research (flexibility and circularity). A theoretical and convenience sampling will be carried out. The composition of the participating cohorts guarantees the heterogeneity of the sample, looking for the presence of the greatest range of the different sociodemographic characteristics (which will serve as a segmentation criterion) to guarantee maximum discursive variability.

### 2.5. Variables

#### 2.5.1. Demographical and Clinical Data

Sex, age, level of education, weight, height, blood pressure, cholesterol, diabetes, tobacco and alcohol use will be considered as a covariates of interest. In addition, we will as well consider psychological or neurocognitive interventions as covariates in our models. Assuming that those patients are taking multiple drugs and experimental treatments, in this case, we will assess the exclusion and inclusion criteria on an individual basis and we will record the different medications and treatments they are taking, as well as their duration.

#### 2.5.2. Clinical Variables

Neurological symptomatology include headache, change in vision, change in hearing, ageusia, anosmia, tremor, fatigue, myalgia (presence or absence of the sign or symptom).Neuropsychiatric symptomatology will be measured using the following tests: Hospital Anxiety Depression Scale (HADS); [64] Geriatric Depression Scale 15-item version (GDS); [65] Pittsburgh Sleep Quality Index (PSQI); [66] Stress Disorder Symptom Severity Scale according to the DSM-V (SDSSS); [67] obsessive-compulsive disorder according to DSM-V (OCD), [67].

#### 2.5.3. Neurocognitive Variables

To measure cognitive performance, all participants will complete an extensive neuropsychological examination with the following tests, which provide measures of multiple cognitive functions. Abstract reasoning and fluid intelligence (Intelligence Vocabulary and Matrix Reasoning, WAIS-III [68]; attention (forward span, WAIS-III [68]; working memory (backward, WAIS-III [68]; visuospatial speed (Symbol Search, WAIS-III; [68] Symbol Coding, WAIS-III [68]; Trail Making Test-A [69]; copy time Rey–Osterrieth Complex Figure [70]; visuospatial and visuoconstructive function (copy accuracy Rey–Osterrieth Complex Figure [70]; verbal memory (total learning and recall-II Rey Auditory Verbal Learning Test [71]; visual memory (memory accuracy, Rey–Osterrieth Complex Figure [70]; language (Boston Naming Test-15 [72]); flexibility (Trail Making Test B-A time); [69] fluency (letter and category fluency); [73] inhibition (interference, Stroop Test [74].Emotional status will be measured using the following tests: the Emotion Regulation Questionnaire [75], which is a 10-item scale designed to measure respondents’ tendency to regulate their emotions in the following two ways: (1) cognitive reappraisal and (2) expressive suppression. Respondents answer each item on a 7-point Likert-type scale ranging from 1 (strongly disagree) to 7 (strongly agree). The Impact of Event Scale-Revised [76] is a self-report questionnaire with 22 questions to capture the DSM-IV criteria for PTSD with a special focus on intrusion, avoidance, hyperarousal, and total subjective stress.

#### 2.5.4. Neuroimaging and Neurophysiological Variables

Brain structural changes will be measured with structural MRI, including volumetry of different anatomical regions, cortical thickness, and structural connectivity of principal white matter tracts.Brain functional changes will be measured with functional MRI, including the activation level of functional networks and whole-brain functional connectivity as measured with GraphVar.For retinal microcirculation disorders, retinography with an amidriatic camera (TRC-NW8) and retinal microvascular analysis (SIRIUS software) to assess the ratio of artery/vessel and the tortuosity will be used.For balance and gait disorders, gait, gait speed, and balance as measured with posturography and dynamometric platform (Dinascan/IBVP600) will be considered.

#### 2.5.5. Inflammatory and Immunology Markers

C-reactive protein.Serum antibodies against SARS-CoV-2 (IgM, IgG (Spike), IgG (Nucleocapsid)).Plasma cytokines IL-6, IL-8, IL-12, TNF-α, IFN-α, MCP-1, TGF-β1, and IL-15 measured Simoa multiplex immunoassay platform and analyzed using HD1 analyzer (Quanterix, Lexington, MA, USA). This technology is sensitive enough to overcome the limitation that some of these cytokines are poorly released in case of impaired functionality and are, therefore, difficult to detect and quantify in plasma. Light plasma neurofilament (NfL) and glial fibrillar acid protein (GFAP) will be measured using Neurology 2-Plex B, 103520.

#### 2.5.6. Lifestyle-Related Variables

The Mediterranean diet assessment tool (PREDIMED) [77] is a brief dietary assessment instrument that consists of 14 short questions whose evaluation aims to offer information about adherence to the Mediterranean diet.For physical activity, the International Physical Activity Questionnaire (IPAQ) [78] will be used. It is a 27-item self-reported measure of physical activity for use with individual adult patients aged 15–69 years old.The Sedentary Behavior Questionary (SBQ) [79] is a summary measure of total sedentary time.Quality of life (EuroQol-5D) [80].For impairment in everyday life, narratives emerging from the 30 semi-structured interviews, identifying specific aspects of the experience and delving into their details, will be used. Contingent repertoires of sociolinguistic resources will be shared within the 4-focus group, allowing participants to give meaning to their experiences.

### 2.6. Statistical Analysis Plan

#### 2.6.1. Quantitative Methodology

##### Data Preprocessing

Raw data will be processed with the appropriate software to extract the variables mentioned in Section 2.5. The data will be reformatted for the analysis and to meet the specific requirements of the statistical packages used in the study. A comprehensive data scrubbing and validation will be conducted to eliminate duplicates, missing data, and treatment indications inconsistent with the diagnosis. The significance level is set at 0.05, and the appropriate corrections for multiple comparisons will be applied. All the analyses will be performed using R version 4.1.3 or higher.

##### Statistical and Qualitative Analysis per Aims

Aims 1–3 include the investigation of the changes in neurocognitive PCC.

This involves analyzing the differences presented by the COVIDPNC group concerning the other three control groups (COVIDPnoN, COVIDR, and NoCOVID). 

For every one of the defined variables, we will define categorical attributes as a function of the results obtained. The variables will be classified as continuous or categorical. Based on this, the corresponding statistical tests will be applied. The baseline descriptive analysis will be conducted using the number and percentage for the categorical variables, the mean and standard deviation for the quantitative variables with normal distribution, and the median and inter-quartile range for the quantitative variables with non-normal distribution. 

In the bivariate proportion comparison, we will use Pearson’s chi-square, and in the continuous variables, we will use the Student’s t-test or Mann–Whitney U test when appropriate. The case–control analysis will be performed with conditional logistic regression analysis and calculate crude and adjusted odds ratio with 95% confidence intervals. 

Aim 4 includes the investigation of the relationship between the changes in neurocognitive PCC.

This involves analyzing the relationship between the neurocognitive status (Section 2.5.3) and the different biomarkers of demographics (Section 2.5.1), clinical status (Section 2.5.2), neurophysiological (Section 2.5.4), inflammation, and the immune response produced by SARS-CoV-2 (Section 2.5.5), and lifestyle (Section 2.5.6).

To estimate the associations between the different variables and events of interest (for cognitive symptoms), the lineal logistic regression conditional models will be adjusted to obtain the OR and the confidence intervals of 95% (CI95%). The models will be adjusted for confusion factors, such as age, sex, previous COVID, comorbidities, tobacco consumption, medication, and sociodemographic data. 

Aim 5 includes describing the impact of persistent neurocognitive symptoms of PCC in daily life through individualized interviews and focus groups.

The scripts of the interviews and the focus groups will be validated using the Delphi method with experts in qualitative research, COVID-19, and primary care. Audio-recorded and video-recorded data from interviews and focus groups will be transcribed within three days of the acquisition. They will be analyzed using Smith’s interpretive phenomenological analysis and the software Atlas.ti [81]. The transcripts of each case will be codified independently. Then, the emerging codes will be harmonized and grouped into intermediate categories to develop a descriptive model of the neurocognitive PCC and its impact on everyday life. Once the data is saturated, the pop-up description will be contrasted with the words of the participants via tri-angulation reviewed by them (member checking).

#### 2.6.2. Integration of Quantitative and Qualitative Approaches

The development of the qualitative and the quantitative studies concurrent and the sequential structuring of the first will allow us to refine the questions of the interviews and obtain a greater depth in aspects relevant to the participants. In the purely analytical phase, the stories obtained in the interviews will provide a coherent and contrasted narrative to expose the neuropsychological symptoms and, in turn, “expose” this symptomatology, detailing how it interferes with or favors specific approaches and confrontations [82].

### 2.7. Ethical Considerations

All the researchers and collaborators adhere to the Helsinki Declaration. Participants will be identified only by a numeric identifier. Only the data controller will have access to the list of the names of the participants, their numeric identifiers, and the date of inclusion. All the information obtained in the study will be treated confidentially, in compliance with Organic Law 3/2018 on protecting personal data and guaranteeing digital rights and the General Data Protection Regulation 2016/679 (GDPR).

## 3. Discussion

### 3.1. Expected Results

The results of this project with the exhaustive tests will allow the identification of biomarkers that will be easy to apply in primary care to identify cases of PCC with neurocognitive impairment (i.e., neurocognitive PCC). In addition, these biomarkers will potentially serve as therapeutic targets for different treatments and trigger similar research to find more definitive solutions for the PCC.

The project will provide essential information on the pandemic for several aspects, which are as follows: (1) it will propose multimodal biomarkers for the different neurocognitive conditions caused by COVID-19, (2) it will be able to identify which structural, functional, psychological, immunological, and inflammatory changes are most prevalent in people with neurocognitive PCC, and (3) it will propose a mechanistic model describing the development and the progression of neurocognitive PCC. 

### 3.2. Impact

The expected findings will significantly impact various areas, such as (a) public health policies, (b) science, (c) economy, and (d) innovation.

#### 3.2.1. Public Health Policies

Improving the current knowledge about the PCC’s neurocognitive conditions will help establish care protocols and clinical practice guidelines, translating into recommendations for diagnostic and intervention strategies and public health policies that policymakers and stakeholders will need to implement.

#### 3.2.2. Scientific Impact

The results of this project will allow us to identify changes in the neural and immunological systems, and the interaction between the two in response to COVID-19. The wide range of tests that will be performed will allow us to identify critical components of neurocognitive PCC and provide a mechanistic understanding of the neurocognitive changes in response to COVID-19.

The results will also serve as a basis for further research with more in-depth studies on the key components and mechanisms to complete our understanding and find solutions to the neuropsychological consequences of PCC.

The multidisciplinary aspect integrating both the neuroscientific and immunological considerations is a crucial element in the study design. Therefore, the communication strategy will be developed to link groups with diverse expertise to be engaged with the project with the most effective means to execute the study and disseminate the findings globally.

#### 3.2.3. Economic Impact

Over 395 million people worldwide have been and are affected by COVID-19, and 10–15% of them suffer from PCC, which indicates a significant impact on the number of working hours. Identifying biomarkers will allow for the development of more efficient rehabilitation strategies to reduce the productive time lost to COVID-19.

Moreover, there is a sustained excessive load on the health services to provide the necessary care to those affected. Having recovered from the initial shock of the pandemic, the longer-term complications of COVID-19 and subsequent demand for healthcare services are starting to affect these services. The health economy that uses both disability-adjusted life years (DALY) and quality-adjusted life years (QALY) can better capture the long-term impact of COVID-19, as up to 30% of the associated health burden may be due to a long-term disability rather than mortality caused by COVID-19 in all age groups [83,84,85]. This prospective study, conducted with a long-term follow-up, will allow the implementation of a longitudinal assessment of health effects and long-term economic impacts of living with PCC. Moreover, this study highlights the importance of investigating the pathogenesis, the risk factors, and the treatment of the many clinical components of PCC in parallel with their effects on society and the health economy [83,84,85].

#### 3.2.4. Innovation

The project proposes an innovative approach to searching for biomarkers of neurocognitive impairment in PCC. On the one hand, the analysis of the retinal microvasculature demonstrates the retina’s role as a window to brain physiology and the state of cerebral vascularization. The retina is the only place in the body where the condition of the macro-and microvasculature can be observed “in vivo” non-invasively and at a low cost. On the other hand, exploring variables such as the speed of the gait and the subtle changes in the postural balance can also relate to alterations in neural functions. These techniques are almost immediately transferable to clinical practice, as most health centers already have them. Moreover, our findings can provide the basis for developing protocols and tools for the automatic and deployable diagnosis of neurocognitive PCC.

### 3.3. Strengths and Limitations

The comprehensive multi-omics design that considers demographic, clinical, neurocognitive, neurophysiological, immunological, and lifestyle aspects combined with mixed methodology provides a holistic view of how PCC affects people’s lives, emphasizing the neurocognitive issues. The multidisciplinary approach integrating neuroscientific and immunological considerations contributes to the advancement of the emerging field of neuroimmunology.

The proposed design with several different control groups delineates primary and secondary causes and mechanisms.

As a limitation, it is possible that COVID-19 vaccines may elicit neurocognitive side-effects in some instances, even in the NoCOVID-19 group. On the other hand, the rate of these side effects is minimal, and the vaccination rate is expected to be comparable across all groups; therefore, its impact will be negligible. 

## 4. Conclusions

The proposed protocol approximates how a comprehensive exploration should be conducted in patients with the post-COVID condition, which presents a neurocognitive profile, to prevent early cognitive dysfunction and perform early treatment in response to this dysfunction. In addition, it offers the opportunity to better understand the pathophysiology behind the symptomatology presented by those patients, allowing for better care approximation. 

## Figures and Tables

**Figure 1 vaccines-10-00849-f001:**
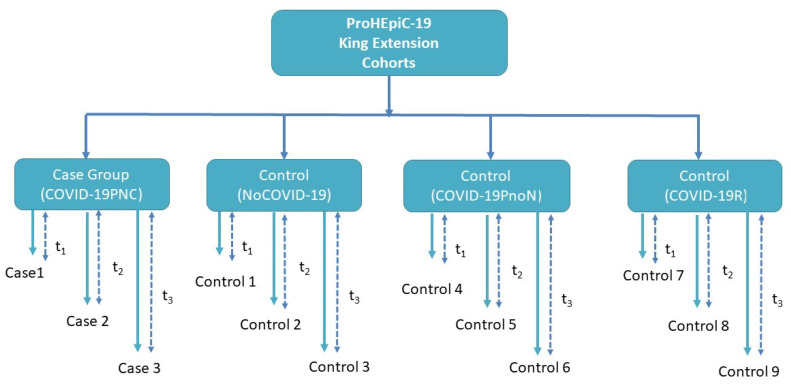
Nested case–control study in a cohort study. Persistent neurocognitive deficits in PCC (COVID19-19PNC) patients. The three control groups include PCC patients with no neurocognitive deficits (COVID-19PnoN), patients who had recovered from COVID-19 (COVID-19R), and healthy controls who did not have COVID-19 (NoCOVID-19).

## Data Availability

Not applicable.

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
