# Peer review of "Neurocognitive Profile of the Post-COVID Condition in Adults in Catalonia—A Mixed Method Prospective Cohort and Nested Case–Control Study: Study Protocol"

_vaccines, 2022, doi:10.3390/vaccines10060849_

Round 1

Reviewer 1 Report

The Authors present a very interesting paper: "Neurocognitive profile of the post-COVID condition in adults in Catalonia: mixed method prospective cohort and nested case-control study" well written and studied. Introduction, Methodology, Statistical Analysis, Results are appropriate and completed, understandable and interesting. Discussion as to me as well as conclusion require only.

  1. a summary table including minimal recommendations
  2. 1. a flow-chart of the way that Authors suggest to follow (essential tests or exams to detect or prevent neuro cognitive degeneration).

Author Response

Dear Rewiew 1

According to the questions considered by the reviewers’ regarding the research article entitled Neurocognitive profile of the post-COVID condition in adults in Catalonia: mixed method prospective cohort and nested case-control study . We are replying to the reviewers' comments, our responses are in blue.

In the event that it is necessary to introduce changes in the text, we indicate them with "track changes"

Point-by-point response

Reviewer reports:

Reviewer 1:

  • English language and style are fine/minor spell check required

Thank you very much for your comment we have now tracked down all the spelling problems in the text.

  • The Authors present a very interesting paper: "Neurocognitive profile of the post-COVID condition in adults in Catalonia: mixed method prospective cohort and nested case-control study" well written and studied. Introduction, Methodology, Statistical Analysis, Results are appropriate and completed, understandable and interesting. Discussion as to me as well as conclusion require only a summary table including minimal recommendations.

Thank you for your comment. We think that at this stage of the research is too early to propose recommendations. However, we certainly have added a conclusion section. We recommend our tests and evaluations in the case of long-covid symptomatology, depending upon the signs and symptoms presented by every patient, although it is not a table as it is too early to draw firm conclusions.

  • A flow-chart of the way that Authors suggest to follow (essential tests or exams to detect or prevent neuro cognitive degeneration). Are the conclusions supported by the results?

Thank you for the question. We still do not have results so we can not state that results are supported or not by the conclusions. We suspect that this will be the case but we are not a hundred percent sure about it. That’s why we are conducting this study.

Reviewer 2 Report

The protocol study "Neurocognitive profile of the post-COVID condition in adults in Catalonia: mixed method prospective cohort and nested case-control study" by Dacosta-Aguayo et al., is a nice and impressive study. The work is well designed and depicted. The methodology employed, results achieved and discussions are well in accordance. The study is timely and of significance. There are just minor grammatical mistakes, like n abstract, there is a word written as "in-ter-views". Please look for similar and other spelling and grammatical mistakes.

Author Response

Dear Reviewer 2:

  • The protocol study "Neurocognitive profile of the post-COVID condition in adults in Catalonia: mixed method prospective cohort and nested case-control study" by Dacosta-Aguayo et al., is a nice and impressive study. The work is well designed and depicted. The methodology employed, results achieved and discussions are well in accordance. The study is timely and of significance. There are just minor grammatical mistakes, like n abstract, there is a word written as "in-ter-views". Please look for similar and other spelling and grammatical mistakes.

Thank you very much for your comments. We have searched for similar and other grammatical mistakes in the whole manuscript. We hope it is now better and it has increased the quality.

Reviewer 3 Report

The manuscript reports a manuscript protocol for a study on the neuropsychological effects of COVID-19 infections. The protocol presented is interesting and well documented. The relevance and the timing of the study are high. However, I have some comments for the authors about their manuscript, looking to be helpful in the increase of their impact:

  • please, include in the title that this is a protocol
  • it is not clear to me how many centers are involved. How many researchers will perform the evaluation? How have you standardized the evaluation across centers and across time points?
  • have you considered the differences between cognitive performances between nearly 18 years old and older people? Control groups seemed very small for this proposal. Moreover, you stated that you are looking for the greatest range of variability, but this aspect will reduce your statistical power, have you considered this aspect in your power analysis? Please justify this aspect. Your major risk is to include too different people with no results due to their heterogeneity. 
  • will you consider medications as covariates or will they be one of the exclusion criteria? Because psychiatric drugs have an effect on cognitive performance and people could start the medication for covid effects
  • why did you include only prior psychiatric conditions as exclusion criteria? what about neurological disorders? 
  • the manuscript needs a revision from a native English speaker for style and checking for typos

Author Response

Reviewer 3:

  • Moderate English changes required

Thank you very much for highlighting this issue. We now have checked the whole manuscript, and we apologize for the mistakes we found in it.

  • The manuscript reports a manuscript protocol for a study on the neuropsychological effects of COVID-19 infections. The protocol presented is interesting and well documented. The relevance and the timing of the study are high. However, I have some comments for the authors about their manuscript, looking to be helpful in the increase of their impact:
  • : please, include in the title that this is a protocol

Thank you for the comment. We have now included the word protocol in the title and now it appears as: “Neurocognitive profile of the post-COVID condition in adults in Catalonia. A mixed method prospective cohort and nested case-control study: Study Protocol”

  • It is not clear to me how many centers are involved. How many researchers will perform the evaluation? How have you standardized the evaluation across centers and across time points?

Thank you very much for including this important comment. We realize we were not enough clear on those points. We have now added that the study will involve the 70 primary care centers in the Northern metropolitan area of Barcelona and the Germans Trias i Pujol University Hospital. The participants will always be evaluated by the same neuropsychologist trained and prepared to perform the different diagnostic tests of the study. The test will be standardized according to every participant's age and years of education. The participants will be all assessed in one center in Mataró. Participants belong to two different cohorts (ProHEpiC_19 and King extension project) to whom we will offer to participate in the new study. They will sign informed consent for the new project.

  • Have you considered the differences between cognitive performances between nearly 18 years old and older people?

Thank you for your interesting comment. As we stated in the previous question, we will have into consideration age and years of education for every participant and we will adjust the different tests for age and years of education to prevent any bias in cognitive performance due to the different of age or education as younger people tend to have more years of education than older ones. 

We also added the age range that is between 25 and 65 years.

  • Control groups seemed very small for this proposal. Moreover, you stated that you are looking for the greatest range of variability, but this aspect will reduce your statistical power, have you considered this aspect in your power analysis? Please justify this aspect. Your major risk is to include too different people with no results due to their heterogeneity.

Many thanks for your consideration. We know that this could be a challenge and therefore 3 different types of controls have been considered, the simple size has enough statistical power as we comment on section 2.4.

Additionally, we clarify the case definition: The persons must have two or more symptoms from points 1 and 2. Two groups of COVID-19PNC will be considered a) persons with 6 to 12 months of symptoms, and persons with more than 12 months of symptoms.

  • Will you consider medications as covariates or will they be one of the exclusion criteria? Because psychiatric drugs have an effect on cognitive performance and people could start the medication for covid effects.

Yes, that is completely true. Thank you for your comment. In a first moment we thought about excluding patients taking psychiatric drugs with effect on cognitive performance but then we realized we were not being realistic as the vast majority of people with post-Covid condition are being treated for several conditions including depression, anxiety…Therefore we have added in the methodology section that we will take into consideration not only the medication prescribed but also the time this medication has been being taken. We will covariate for them.

We added neurological disorders previous to COVID-19 as exclusion criteria.

  • Why did you include only prior psychiatric conditions as exclusion criteria? what about neurological disorders?

Yes, you are totally right. We have specified severe psychiatric conditions, neurological diseases and history of drug or alcohol abuse prior to covid infection as exclusion criteria.

The manuscript needs a revision from a native English speaker for style and checking for typos.

Thank you for your comment, we have now improved the quality of the writing and we have looked for all the typos.  The whole manuscript was revised by a neuroscientific with the highest ( native ) competence in the English language.

Round 2

Reviewer 3 Report

I think the authors have addressed all my concerns. I think now the manuscript could be considered for publication